# Association between Tongue Pressure and Jaw-Opening Force in Older Adults

**DOI:** 10.3390/ijerph19169825

**Published:** 2022-08-09

**Authors:** Chizuru Namiki, Koji Hara, Ryosuke Yanagida, Kazuharu Nakagawa, Kohei Yamaguchi, Takuma Okumura, Tomoe Tamai, Yukiko Kurosawa, Tomoko Komatsu, Haruka Tohara

**Affiliations:** 1Department of Critical Care Medicine and Dentistry, Graduate School of Dentistry, Kanagawa Dental University, Yokosuka 2388580, Japan; 2Department of Dysphagia Rehabilitation, Graduate School of Medical and Dental Sciences, Tokyo Medical and Dental University, Tokyo 1138510, Japan

**Keywords:** sarcopenia, tongue pressure, jaw-opening force, geniohyoid muscle, suprahyoid muscle

## Abstract

Tongue pressure (TP) is used to assess tongue muscle strength and is related to function and frailty. While performing TP, it is necessary to elevate the tongue and oral floor by contracting the suprahyoid muscles. However, the association between TP and suprahyoid muscle strength remains unclear. Accordingly, this study investigated the relationship between TP and jaw-opening force (JOF), an indicator of suprahyoid muscle strength. This cross-sectional study included 88 independent community-dwelling participants aged ≥65 years. Age, sex, and the number of remaining teeth were recorded. Ultrasonography was used to evaluate the cross-sectional area of the tongue and geniohyoid muscle, as representatives of the suprahyoid muscles. Sarcopenia was diagnosed based on appendicular skeletal muscle mass index, handgrip strength, and gait speed. Multiple regression analysis was performed with TP as the dependent variable. TP was significantly associated with JOF (β = 0.371, *p* = 0.003). This study revealed that decreased TP was associated with a decline in JOF and suprahyoid muscle mass in older adults. Thus, low TP may be associated with decreased JOF. Prevention of the weakness of the suprahyoid muscles and maintaining TP may also contribute to the prevention of frailty associated with TP.

## 1. Introduction

The tongue significantly contributes to bolus formation and manipulation, as well as to the safe transportation of food from the oral cavity into the pharynx. Additionally, to transport the bolus into the esophagus, pharyngeal swallowing depends on the base of the tongue moving and contacting the posterior wall of the pharynx [1]. Therefore, weakness of the tongue muscles may cause dysphagia. Tongue pressure (TP) is an indicator of tongue muscle strength [2], and decreased TP is associated with poor bolus formation and choking while eating [3]. There is a relationship between dysphagia and pharyngeal residuals during swallowing [4]. The typical values of TP have been established for different age groups [5], with TP decreasing with age [6]. Additionally, TP decreases with dysphagia-causing diseases such as cerebrovascular disease [7], Parkinson’s disease [8], and sarcopenia [6]. TP is one of the important indicators of oral function. Thus, it is crucial for community-dwelling older adults to maintain healthy oral functions, to prevent frailty throughout life [9]. Notably, TP is associated with physical functions such as handgrip strength (HGS) in community-dwelling older adults [10], and decreased TP is a high-risk factor for frailty in older adults [9]. Therefore, it is essential to identify the factors associated with low TP in community-dwelling older adults. A recent report showed a relationship between the cross-sectional area (CSA) of the geniohyoid (GH) muscle and TP in participants with sarcopenic dysphagia [11]; this relationship can be explained by the mechanism of TP generation. TP is generated by lifting the tongue and elevating the mouth floor when the suprahyoid muscle contracts [12]. Therefore, when the strength of the suprahyoid muscle decreases, TP may decrease. However, there have been no reports on the relationship between TP and suprahyoid muscle strength. As the suprahyoid muscle is a jaw-opening muscle, a jaw-opening stenometer has been developed to evaluate the strength of the suprahyoid muscle [13]. Previous studies have shown that jaw-opening force (JOF) is associated with dysphagia in patients with dysphagia [13,14]. Furthermore, in older adults, JOF is associated with the resting position of the hyoid bone [15] and CSA of the GH muscle [16], suggesting that opening force may be a useful indicator of suprahyoid muscle strength. Although TP and JOF have a strong relationship, to our knowledge, there have been no reports on their relationship until now. In this study, we investigated the relationship between TP and JOF, an index of suprahyoid muscle strength. Considering that suprahyoid muscle contraction plays an important role when conducting TP, we hypothesized that decreased TP in older adults is associated with suprahyoid muscle strength. The aim of this cross-sectional study was to clarify the association between TP and JOF as an index of suprahyoid muscle strength in older adults.

## 2. Materials and Methods

### 2.1. Ethics Statement

The study was approved by the Ethics Committee of the School of Dentistry, Tokyo and Medical Dental University (D2014-047). Oral and written informed consent was obtained from all participants. All experiments were performed in accordance with the relevant guidelines and regulations. This study was also conducted in accordance with the latest revision of the Declaration of Helsinki. The authors obtained written informed consent to publish images of the patients in Figure 1, Figure 2 and Figure 3.

### 2.2. Participants

A total of 122 independent community-dwelling participants aged ≥65 years were recruited for this cross-sectional study in October 2018 in Oyama, Tochigi, Japan. Recruitment was announced by the Oyama City Hall and the Oyama City Dental Association (Tochigi, Japan). The final analysis was based on 88 community-dwelling participants (30 men, 58 women; mean age, 71.0 ± 5.4 years). The inclusion criteria were as follows: (i) age >65 years; (ii) ability to follow instructions for measurements of jaw-opening, HGS, and gait speed; and (iii) provision of informed consent to participate in this study. The exclusion criteria for this study were as follows: (i) poorly controlled disease symptoms, (ii) a history of oral or pharyngeal surgery, or (iii) dementia and other conditions that make it difficult to follow instructions.

### 2.3. Procedures

#### 2.3.1. Data Collection

We collected data based on the medical records of the participants. Age, sex, height, and weight were recorded. Body mass index (BMI) was calculated as weight (kg)/height (m^2^). The number of remaining teeth and denture use were evaluated visually during an oral examination, by dentists with >5 years of experience, and classified using the Eichner classification.

#### 2.3.2. TP

A JMS TP device (JMS Co., Ltd., Hiroshima, Japan) (Figure 1) was used to measure TP. Participants were placed in a seated position in a chair.

The participants were instructed to place the balloon on their tongue, close their lips, and hold a plastic pipe with their upper and lower central incisors. After confirming the correct position, the dentist grasped the plastic probe and performed the measurements. The participants were instructed to press their tongue against their palate for 7 s [5]. An interval between measurements, of approximately 30 s, was set during the measurement. The maximum TP was defined as the average of three measurements.

#### 2.3.3. JOF

Based on the method outlined in a previous study, a jaw-opening stenometer [13] (Livet Inc., Tokyo, Japan) was used to measure JOF (Figure 2).

Measurements were performed with the participants sitting in a chair. Before the measurement, the participants were instructed to bite down on their teeth together. The JOF was measured accurately by placing an adjustment belt on top of the head, and a chin guard was placed under the chin to hold it as tightly as possible [13]. The participants were then asked to open their jaws as wide as possible. The JOF was measured for 5 s, the belt was retightened at every measurement, and an interval of approximately 30 s was set during measurement. The JOF was defined as the average of three measurements [13]. Blinding was employed for both JOF and TP measurements.

#### 2.3.4. Assessment of Sarcopenia

Sarcopenia was diagnosed according to the 2019 Asian Working Group for Sarcopenia criteria [17]. HGS, gait speed, and bioelectrical impedance analysis (BIA) were used to assess muscle strength, physical function, and skeletal muscle mass, respectively. Muscle strength was measured using a handgrip dynamometer (Matsuyoshi & Co., Ltd., Tokyo, Japan), with the participant standing. HGS was measured twice using the dominant hand, and the average of the two measurements was used in the analysis. Low HGS was defined as <28 kg for men and <18 kg for women [17]. Physical function was assessed based on gait speed. Low gait speed was defined as <1 m/s. BIA was performed using an Inbody S10 body composition analyzer (Inbody Japan, Inc., Tokyo, Japan), and the measurements were taken in a relaxed sitting position. Appendicular skeletal muscle mass index (ASMI) measurements were adjusted by dividing ASM by the square of height (m). The BIA cutoff for low muscle mass was set at <7.0 kg/m^2^ for men and <5.7 kg/m^2^ for women [17]. Based on these parameters, sarcopenia was defined as low muscle mass, weaker HGS, and/or slow gait speed.

#### 2.3.5. Muscle Mass of the Tongue and Suprahyoid Muscle

For the evaluation of the tongue and suprahyoid muscle mass, an ultrasound material (SonoSite M Turbo; Fujifilm, Tokyo, Japan) with a 5–15 MHz was used to measure the area of the tongue and suprahyoid muscle mass using a convex-array transducer. Participants were seated in a chair without a backrest, with their back straight. We instructed the participants to close their mouth lightly. The number of measurements and transducer positions were consistent with previous studies [11,18,19]. To assess the area of the hyoid muscle in the coronal plane, the probe was placed on a horizontal line connecting the parotid gland to the mandible at 1/3 of the distance. In the suprahyoid muscle group, the GH muscle was selected based on a previous study [11]. The area of the GH muscle in the sagittal plane was measured by placing the transducer on the midline of the floor of the mouth, perpendicular to the horizontal plane. Parameters were calculated using ImageJ software (National Institutes of Health, Bethesda, MD, USA), and an average of three replicates was used for analysis (Figure 3).

To assess the reliability of the ultrasonography, we calculated interclass correlation coefficients. The intra- and inter-rater reliabilities of tongue muscle mass were 0.814 and 0.969, respectively, and the intra- and inter-rater reliabilities of the GH muscle mass were 0.964 and 0.875, respectively.

### 2.4. Statistical Analysis

A Shapiro–Wilk test determined whether the data conformed to a normal distribution. An unpaired t-test and the Mann–Whitney U-test analyzed parametric and nonparametric data, respectively. A chi-square test was used to compare nominal data between the groups with and without sarcopenia and between men and women. For multiple regression analysis, explanatory variables including age and sex were selected, based on whether the variables were related to TP, according to previous reports [1] (sarcopenia [6], tongue muscle area [11,18], GH muscle area [11], and remaining number of teeth [9]). Multiple regression analysis was performed with TP as the dependent variable and age, sex, muscle mass index, JOF, and tongue muscle area as independent variables (model 1). In addition, as the correlation coefficient between JOF and the CSA of the GH muscle was high (r = 0.649), a multiple regression analysis was performed with the CSA of the GH muscle as an independent variable instead of JOF (model 2), to avoid multicollinearity. In all analyses, if the variance inflation coefficient was <3.0, and the significance level was set at *p* < 0.05 (effect size, 0.15 (medium); power, 0.8; the number of predictors, 6; α = 0.05), then the multivariate linear analysis was not affected by the multicollinearity. The Japanese version of SPSS for Windows (version 25 J, IBM Japan Co., Ltd., Tokyo, Japan) was used for all statistical analyses.

#### Sample Size Calculation

We calculated the required sample size using G* Power 3.1 (Heinrich-Heine-Universität Düsseldorf, Düsseldorf, Germany). According to a previous report [11], we calculated the effect size of the association between TP and GH muscle, which was set to 0.38. According to our calculations (α = 0.05, power = 0.80, and effect size = 0.38), 43 participants were required for this study.

## 3. Results

We excluded 31 participants with missing data and three with poor ultrasound image quality. Among these, the jaw-opening stenometer malfunctioned for 27 patients. Seven patients had missing data about remaining teeth and ultrasound images. Of the 88 participants, two men (2.3%) and five women (5.7%) were diagnosed with sarcopenia. Participant characteristics are listed in Table 1, and Table 2 shows the characteristics for men and women, while Table 3 shows those of the sarcopenia and non-sarcopenia groups. There were significant sex differences in age (*p* = 0.003), JOF (*p* < 0.001), CSA of the GH muscle (*p* < 0.001), and tongue muscle area (*p* = 0.013).

Moreover, there were significant sex differences in HGS (*p* < 0.001) and ASMI (*p* < 0.001), which are indicators of sarcopenia. However, there were no significant between-sex differences in the incidence of sarcopenia (Table 2). Moreover, there were significant age differences (*p* = 0.015), JOF (*p* = 0.011) and CSA of the GH muscle (*p* = 0.049) between the sarcopenic and non-sarcopenic groups (Table 3). The correlations for each item are presented in Table 4.

TP was significantly correlated with ASMI (r = −0.271, *p* < 0.05), JOF (r = 0.310, *p* < 0.05), and CSA of the GH muscle (r = 0.303, *p* < 0.05), but not with tongue muscle area (r = 0.008, *p* = 0.415), number of remaining teeth (r = −0.086, *p* = 0.425), or age (r = −0.137, *p* = 0.202).

Table 5 shows the results of the multivariate analysis of TP (model 1 and model 2). After adjusting for sex and age, TP was significantly associated with JOF (β = 0.378, *p* = 0.003), but not with age (β = −0.013, *p* = 0.918), sex (β = 0.201, *p* = 0.118), sarcopenia (β = 0.162, *p* = 0.137), tongue muscle area (β = 0.008, *p* = 0.943), or number of remaining teeth (β = −0.008, *p* = 0.464) (model 1). In addition, the CSA of the GH muscle (β = 0.408, *p* = 0.003) was also a significant explanatory variable for TP (model 2). However, it was not associated with age (β = −0.050, *p* = 0.680), sex (β = 0.252, *p* = 0.067), sarcopenia (β = 0.156, *p* = 0.151), tongue muscle area (β = −0.037, *p* = 0.736), or number of remaining teeth (β = −0.044, *p* = 0.687).

## 4. Discussion

Our study found that TP was independently correlated with the JOF and CSA of the GH muscle. TP is generated by lifting the tongue itself and elevating the mouth floor, which occurs due to hyoid elevation caused by the contraction of the suprahyoid muscle [12]. The muscles involved in TP generation include the mylohyoid, anterior belly of the digastric, and GH muscles of the suprahyoid muscle [19]. A surface electromyogram study showed that TP was correlated with suprahyoid muscle activity during TP resistance training [20]. Our previous study found that TP resistance training in patients with presbyphagia improved the anterior elevation of the hyoid bone, the amount of upward elevation, and TP [21]. Notably, some suprahyoid muscles are involved in jaw-opening, and these muscles include the GH muscle, the anterior belly of the digastric muscles, and the mylohyoid muscle [19]. Considering these previous findings, the association between TP and JOF demonstrated in this study is reasonable. In addition, Mori et al. showed an association between TP and muscle mass of the superior hyoid muscle and CSA of the GH muscle, which was measured using an ultrasound device in patients with sarcopenic dysphagia [11].

Consistently, in this study, the CSA of the GH muscle was a significant explanatory factor for TP. The associations of TP with the CSA of the GH muscle and JOF strongly suggest that the suprahyoid muscle is significantly related to TP, which plays a role in jaw-opening. Moreover, because the anterior belly of the digastric and the mylohyoid muscles are involved in the JOF, further studies are needed to investigate the association between TP and CSA of these muscles. Mori et al. showed no significant correlations between the CSA of the GH muscle, TP, and tongue muscle area [11]. In the present study, we consistently measured the CSA of the tongue using an ultrasound device. We performed multivariate analyses, and the results showed no relationship between TP and the tongue muscle area. Accumulating studies have indicated that TP declines with age [6]; the tongue muscle area, however, is positively correlated with aging and tends to increase with age [22].

Moreover, it has been stated that aging of the tongue is associated with fatty deposition in the tongue [22]. Although tongue fat was not measured in this study, the results support these findings. However, the tongue muscle area and TP were measured by ultrasonography and showed a decrease in patients with sarcopenic dysphagia [6,18]. Therefore, although this association remains controversial, the relationship between TP and tongue muscle area may be significant in older adults with sarcopenia. Notably, a decrease in TP has been associated with severe sarcopenia [6]. However, TP was not associated with sarcopenia in the present study. There are two possible explanations for this inconsistent finding. One reason is the difference in the prevalence of sarcopenia among the participants. In a previous study, which suggested an association between TP and sarcopenia, 34.5% of older adults were diagnosed with sarcopenia [6].

In contrast, the number of participants diagnosed with sarcopenia was small (7 of 88; 8%) in this study. Based on the 2019 Asian Working Group for Sarcopenia criteria, the incidence of sarcopenia is reported to be 2.9% at 65 years of age in Japan [23]. Based on this fact, the sarcopenia incidence rate in this study was considered reasonable. Another factor is the difference in the criteria used for sarcopenia. In a previous study, which suggested an association between TP and sarcopenia, the criteria were low grip strengths of <26 kg for men and <18 kg for women [24].

In contrast, the diagnostic criteria for HGS for sarcopenia in this study were low grip strengths of <28 kg for men and <18 kg for women [17]. There was a difference in the cutoff value for HGS. Consequently, TP was not associated with sarcopenia in this study.

In this study, a multiple regression analysis showed no positive association between TP and the number of remaining teeth, but a negative correlation was found. Hara et al. reported that tooth loss could strengthen the TP in older adults; in other words, increased TP may be compensating for tooth loss to maintain masticatory function [25]. However, it has been reported that the number of remaining teeth decreases with reduced TP in frail patients [9]. Therefore, further studies are needed to investigate the association between the tongue and the number of remaining teeth in patients with sarcopenia and frailty.

The association between TP and JOF may indicate that dynapenia, which decreases muscle strength without a change in muscle mass [26], in the suprahyoid muscle reduces TP. The decline in muscle strength is generally related to muscle mass loss. Nevertheless, it may also be related to neuromuscular factors (such as neuromuscular junction alterations) and loss of muscle mass due to neural factors. Electromyographic studies have shown that the muscle strength gains seen in the early stages of resistance training in older adults are due to improvements in neural factors, rather than muscle hypertrophy [27,28]. Therefore, clinicians should pay attention to the loss of suprahyoid muscle and the strength of the suprahyoid muscles in clinical practice. TP may decrease if dynapenia [26] occurs in the suprahyoid muscle.

Furthermore, training to strengthen the tongue muscles and suprahyoid muscles, such as with the Shaker exercise [29] and with the jaw-opening exercises [30], may be effective in improving TP. For patients with difficulties in tongue muscle strength training due to hypoglossal nerve paralysis, Shaker and jaw-opening exercises are expected to improve TP. Interventions for strengthening the suprahyoid muscle to improve the TP should be investigated in the future.

Our study had some limitations. First, the participants in this study were healthy older adults. It will be necessary to investigate the association between the degree of dysphagia, the number of remaining teeth, and TP and JOF using swallowing video endoscopy or videofluoroscopy in patients with dysphagia.

Second, although this was a cross-sectional study, training to strengthen the tongue muscles and suprahyoid muscles, such as with the Shaker exercise [29] and with the jaw-opening exercises [30], may be effective in improving TP. Interventions for strengthening the suprahyoid muscle to improve the TP should be investigated in the future.

## 5. Conclusions

TP was an independent explanatory factor for JOF and geniohyoid muscle mass. This study suggests that decreased TP in healthy older adults may be associated with decreased suprahyoid muscle strength and mass, which might cause a decline in TP. Therefore, exercise to strengthen the suprahyoid muscles may be effective for the decreased TP in older adults. In addition, preventing muscle weakness of the suprahyoid muscles and maintaining TP may contribute to the prevention of frailty, which is related to TP.

## Figures and Tables

**Figure 1 ijerph-19-09825-f001:**
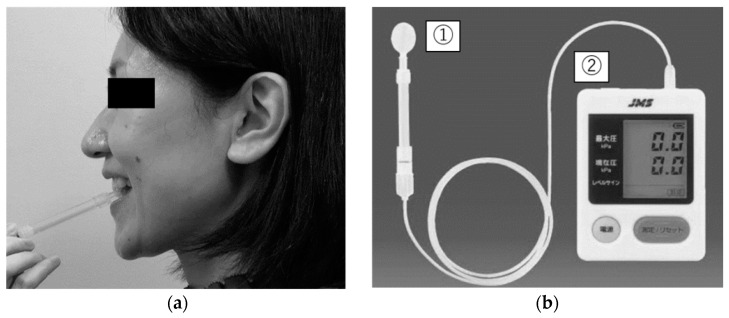
Tongue pressure measurement (Tongue pressure device; JMS Co., Ltd., Hiroshima, Japan). (**a**) Balloon probe; (**b**) Tongue pressure device.

**Figure 2 ijerph-19-09825-f002:**
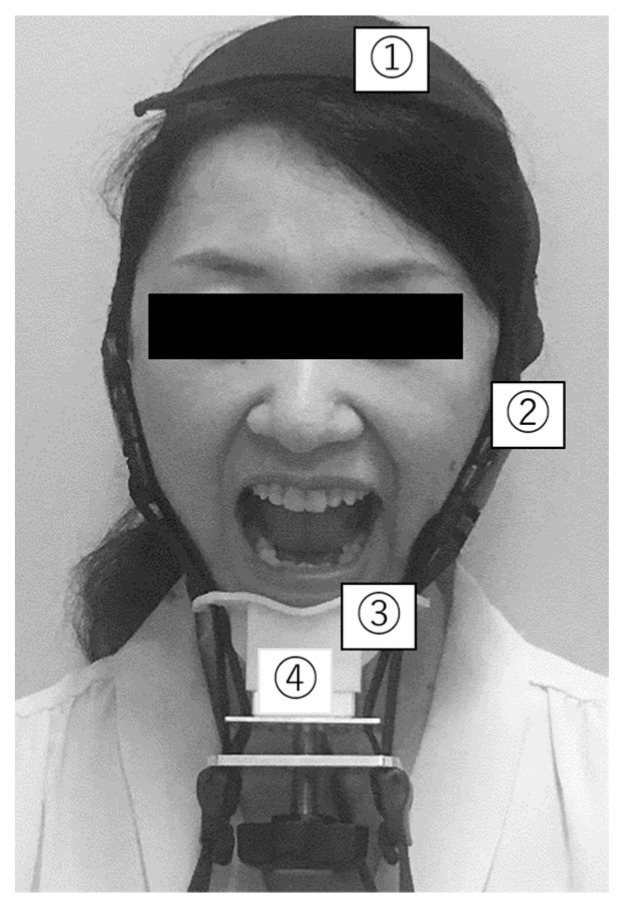
Jaw-opening force measurement (jaw-opening stenometer). ① Head-encircling belt; ② Two straps to secure the mandible to the head-encircling belt; ③ Chin cap; ④ Dynamometer to assess jaw-opening strength.

**Figure 3 ijerph-19-09825-f003:**
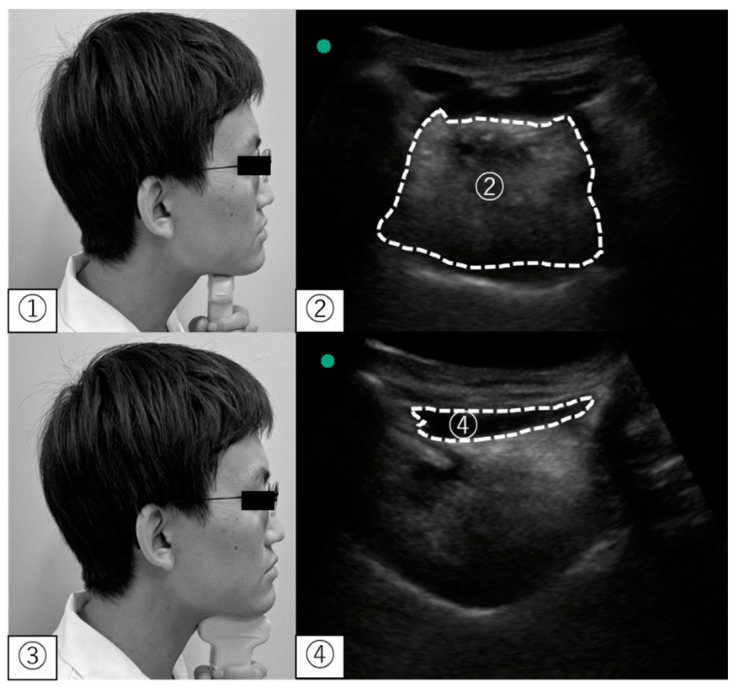
Ultrasonographic measurement. ① Position of the convex probe for the tongue; ② The cross-sectional area of the tongue in an ultrasonographic image; ③ Position of the convex probe for the geniohyoid muscle; ④ The cross-sectional area of the geniohyoid muscle in ultrasonographic images.

**Table 1 ijerph-19-09825-t001:** Characteristics of participants.

Variables	Total (*n* = 88)
Age, years, median (IQR)	70.0 (67.0–74.8)
BMI, kg/m^2^, median (IQR)	22.6 (21.4–24.4)
Tongue pressure, kPa, median (IQR)	34.2 (27.8–37.7)
Jaw-opening force, kg, median (IQR)	6.6 (5.3–8.1)
Sarcopenia, *n* (%)	7 (8.0)
ASMI, kg/m^2^, median (IQR)	6.3 (5.8–7.2)
Handgrip strength, kg, median (IQR)	24.9 (21.1–33.2)
Low gait speed, m/s, median (IQR)	1.4 (1.3–1.6)
Tongue muscle area, mm^2^, median (IQR)	1596.4 ± 274.8
Geniohyoid muscle area (sagittal) mm^2^, mean ± SD	280.7 (236.2–321.0)
Eichner classification *n*	
1	62
2	17
3	9
Number of teeth (IQR)	25 (23–28)

BMI: body mass index, ASMI: appendicular skeletal muscle mass index. The numbers refer to the Eichner classification: 1, Eichner A; 2, Eichner B; 3, Eichner C. IQR: interquartile range, SD: standard deviation.

**Table 2 ijerph-19-09825-t002:** Comparisons of characteristics of men and women.

Variables	Male (*n* = 30)	Female (*n* = 58)	*p*-Value
Age, years, median (IQR) ‡	74.0 (70.0–76.0)	69.0(66.8–72.3)	0.003 **
BMI, kg/m^2^, median (IQR) ‡	22.6 (21.8–24.0)	22.6 (21.1–24.5)	0.718
Tongue pressure, kPa, median (IQR) ‡	34.6 (27.3–38.8)	33.8 (27.8–37.6)	0.748
Jaw-opening force, kg, median (IQR) ‡	7.9 (6.2–10.6)	5.9 (5.0–7.3)	<0.001 **
Sarcopenia, *n* (%) ⁑	2 (6.7)	5 (8.6)	0.748
ASMI, kg/m^2^, median (IQR) ‡	7.32 (6.57–7.71)	6.13 (5.54.−6.41)	<0.001 **
Handgrip strength, kg, median (IQR) ‡	34.4 (29.0–39.0)	22.8 (19.5–25.3)	<0.001 **
Low gait speed, m/s, median (IQR) ‡	1.41 (1.25–1.55)	1.45 (1.25–1.6)	0.326
Tongue muscle area, mm^2^, median (IQR) ‡	1636.6 (1451.9–1926.5)	1497.7 (1375.0–1740.8)	0.013 *
Geniohyoid muscle area (sagittal) mm^2^, mean ± SD †	329.1 ± 87.0	253.6 ± 49.7	<0.001 **
Eichner classification, *n*			
1	27	35	
2	1	16	
3	2	7	
Number of teeth (IQR)	27 (24–28)	25 (21–28)	0.239

BMI: body mass index, ASMI: appendicular skeletal muscle mass index. The numbers refer to the Eichner classification: 1, Eichner A; 2, Eichner B; 3, Eichner C. IQR: interquartile range, SD: standard deviation. * *p* < 0.05; ** *p* < 0.01. † *t*-test; ‡ Mann–Whitney U-test; ⁑ Chi-square test. Cutoffs for low muscle mass: <7.0 kg/m^2^ in men and <5.7 kg/m^2^ in women.

**Table 3 ijerph-19-09825-t003:** Comparisons of characteristics between sarcopenia and non-sarcopenia groups.

Variables	Sarcopenia (*n* = 7)	Non-Sarcopenia (*n* = 81)	*p*-Value
Age, years, median (IQR) ‡	76.0 (70.0–81.0)	70.0 (67.0–74.0)	0.015 *
Sex, male, n (%) ⁑	2 (28.5%)	28 (34.6%)	0.748
Sex, female, n (%) ⁑	5 (71.4%)	53 (65.4%)	data
BMI, kg/m^2^, median (IQR) ‡	21.4 (20.0–22.6)	22.6(21.5–24.4)	0.094
Tongue pressure, kPa, median (IQR) ‡	29.8 (15.9–36.8)	34.3 (28.0–37.8)	0.141
Jaw-opening force, kg, median (IQR) ‡	4.5 (3.8–5.2)	6.8 (5.5–8.1)	0.011 *
ASMI, kg/m^2^, median (IQR) ‡	5.5 (5.0–5.7)	6.4 (6.00–8.1)	<0.001 **
Handgrip strength, kg, median (IQR)	17.9 (16.6–27.1)	25.3 (22.2–33.7)	0.01 *
Low gait speed, m/s, median (IQR) ‡	1.2 (1.0–1.4)	1.5(1.3–1.6)	0.018 *
Tongue muscle area, mm^2^, median (IQR) ‡	1440.6 (1386.3–1642.5)	1567.6 (1388.6–1815.7)	0.343
Geniohyoid muscle area, sagittal, mm^2^, mean ± SD †	241.5 ± 67.0	285.8 ± 68.7	0.049 *
Eichner classification, *n*			
1	4	58	
2	2	15	
3	1	8	
Number of teeth (IQR)			
	25(19–26)	25(21–28)	0.530

* *p* < 0.05; ** *p* < 0.001. † *t*-test; ‡ Mann–Whitney U-test; ⁑ Chi-square test. Cutoffs for low muscle mass: <7.0 kg/m^2^ in men and <5.7 kg/m^2^ in women. BMI: body mass index; ASMI: appendicular skeletal muscle mass index; The numbers refer to the Eichner classification: 1, Eichner A; 2, Eichner B; 3, Eichner C. IQR: interquartile range; SD: standard deviation.

**Table 4 ijerph-19-09825-t004:** Spearman correlation coefficient analysis results of the correlations among study variables.

Variables	Age	Tongue Pressure	Jaw-Opening Force	Tongue Muscle Area	Geniohyoid Muscle Area	Number of Teeth	Eichner Classification	ASMI	Handgrip Strength
Age	1								
Tongue pressure	−0.137	1							
Jaw-opening force	−0.196	0.310 **	1						
Tongue muscle area,	0.035	0.088	0.197	1					
Geniohyoid muscle area,	−0.044	0.303 **	0.649 **	0.305 **	1				
Number of teeth	−0.229 *	−0.086	0.123	0.075	0.067	1			
Eichner classification	0.083	0.156	−0.077	−0.101	−0.144	−0.714 **	1		
ASMI	0.016	−0.271	0.510 **	0.354 **	0.480 **	0.160	−0.209	1	
Hand grip strength	−0.007	0.191	0.643 **	0.218 *	0.606 **	0.149	−0.217 *	0.64 **	1
Gait speed	−0.230 *	0.105	0.215 *	−0.034	0.174	0.112	−0.055	0.100	0.143

* *p* < 0.05; ** *p* < 0.01. B: partial regression coefficient, β: standardized partial regression coefficient. VIF: variance inflation factor, ASMI: appendicular skeletal muscle mass index.

**Table 5 ijerph-19-09825-t005:** Multivariate linear regression analysis of factors for tongue pressure.

	Independent Variable	B	β	95% Confidence Interval	VIF	*p*-Value	Adjust R^2^
**Model 1** Tongue pressure (Jaw-opening Force)	Age	−0.017	−0.013	−0.345 to 0.311	1.481	0.918	0.101
	Sex	3.028	0.201	−0.789 to 6.844	1.576	0.118	
	Tongue muscle area	0.000	0.008	−0.005 to 0.006	1.112	0.943	
	Sarcopenia	4.263	0.162	−1.308 to 9.906	1.123	0.137	
	Jaw-opening force	1.155	0.378	0.397 to 1.912	1.445	0.003 **	
	Number of teeth	−1.396	−0.008	−0.345 to 0.159	1.422	0.464	
**Model 2** Tongue pressure (Geniohyoid muscle area)	Age	−0.066	−0.050	−0.386 to 0.253	1.410	0.680	0.104
	Sex	3.781	0.252	−0.277 to 7.839	1.787	0.067	
	Tongue muscle area	−0.001	−0.037	−0.007 to 0.005	1.172	0.736	
	Sarcopenia	4.110	0.156	−1.535 to 9.756	1.127	0.151	
	Geniohyoid muscle area	0.042	0.408	0.015 to 0.069	1.712	0.003 **	
	Number of teeth	−0.051	−0.044	−0.305 to 0.202	1.171	0.687	

Notes: ** *p* < 0.01. Abbreviations: B, partial regression coefficient; β, standardized partial regression coefficient; VIF, variance inflation factor.

## Data Availability

The data that support the findings of this study are available from the corresponding author, Koji Hara, upon reasonable request.

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
