# Peer review of "Association between Tongue Pressure and Jaw-Opening Force in Older Adults"

_ijerph, 2022, doi:10.3390/ijerph19169825_

Round 1

Reviewer 1 Report

Manuscript ID: ijerph-1793950

in the paper submitted for review, authors showed that the cross-sectional area (CSA) of the geniohyoid (GH) muscle was a significant explanatory factor for tongue pressure (TP). Decreased TP was associated with decline in  the jaw-opening force (JOF) and suprahyoid muscle mass in older adults. Finally authors suggest to improve TP to strengthen the suprahyoid muscles as well as tongue muscles.

 The authors used tongue pressure device, a jaw-opening stenometer, a handgrip dynamometer, an Inbody S10 body composition analyzer, which is an advantage of the paper to achieve the aims of the study.

The statistical methods were chosen correctly and the results obtained are not questionable. The results were clearly presented in tables.

The results in the Discussion section have been correctly referred to the current world knowledge.

Only one suggestion:

line 125: please add whether the height squared was in meters or in centimeters.

Reviewer 2 Report

Title:  it would be better if it were shorter and more incisive Abstract: must be rewritten Introduction : the purpose is not clear

2.3.2. Assessment of sarcopenia : must be well explained why it was insered

2.3. Procedures  : need to be better explained

Conclusions : they must be expanded as they are insufficient

Reviewer 3 Report

This is a cross-sectional study examining the association between tongue pressure, suprahyoid muscle strength, and suprahyoid muscle mass in older adults. 

I think the methodology is overall solid; however, the inclusion of only healthy adults make it difficult to draw conclusion regarding the impact on overall health.

1. Besides the "number of teeth" remaining, did the authors investigate other dental parameters such as how many occluding pairs or whether the participants wear removable dentures?

2. Line 106 in methodology: the word "impede" is not correct. I am assuming the authors meant that the participants were instructed to bite down on their teeth together?
